REPLICATION STUDY

# Replication Study: Coding-independent regulation of the tumor suppressor PTEN by competing endogenous mRNAs

Hongyan Wang[1†], Hanna S Radomska[1], Mitch A Phelps[1], Reproducibility Project: Cancer Biology[2,3]*

[1]Pharmacoanalytic Shared Resource (PhASR), Comprehensive Cancer Center, The Ohio State University, Columbus, United States; [2]Science Exchange, Palo Alto, United States; [3]Center for Open Science, Charlottesville, United States

*For correspondence: nicole@scienceexchange.com; tim@cos.io

Present address: [†]Department of Pathology, Duke University School of Medicine, Durham, United States

Group author details: Reproducibility Project: Cancer Biology See page 16

**Abstract** As part of the Reproducibility Project: Cancer Biology, we published a Registered Report (Phelps et al., 2016) that described how we intended to replicate selected experiments from the paper 'Coding-independent regulation of the tumor suppressor PTEN by competing endogenous mRNAs' (Tay et al., 2011). Here, we report the results. We found depletion of putative PTEN competing endogenous mRNAs (ceRNAs) in DU145 cells did not impact *PTEN* 3'UTR regulation using a reporter, while the original study reported decreased activity when *SERINC1*, *VAPA*, and *CNOT6L* were depleted (Figure 3C; Tay et al., 2011). Using the same reporter, we found decreased activity when ceRNA 3'UTRs were overexpressed, while the original study reported increased activity (Figure 3D; Tay et al., 2011). In HCT116 cells, ceRNA depletion resulted in decreased PTEN protein levels, a result similar to the findings reported in the original study (Figure 3G,H; Tay et al., 2011); however, while the original study reported an attenuated ceRNA effect in microRNA deficient (Dicer[Ex5]) HCT116 cells, we observed increased PTEN protein levels. Further, we found depletion of the ceRNAs *VAPA* or *CNOT6L* did not statistically impact DU145, wild-type HCT116, or Dicer[Ex5] HCT116 cell proliferation. The original study reported increased DU145 and wild-type HCT116 cell proliferation when these ceRNAs were depleted, which was attenuated in the Dicer[Ex5] HCT116 cells (Figure 5B; Tay et al., 2011). Differences between the original study and this replication attempt, such as variance between biological repeats, are factors that might have influenced the results. Finally, we report meta-analyses for each result.

## Introduction

The Reproducibility Project: Cancer Biology (RP:CB) is a collaboration between the Center for Open Science and Science Exchange that seeks to address concerns about reproducibility in scientific research by conducting replications of selected experiments from a number of high-profile papers in the field of cancer biology (*Errington et al., 2014*). For each of these papers, a Registered Report detailing the proposed experimental designs and protocols for the replications was peer reviewed and published prior to data collection. The present paper is a Replication Study that reports the results of the replication experiments detailed in the Registered Report (*Phelps et al., 2016*) for a paper by *Tay et al., 2011* and uses a number of approaches to compare the outcomes of the original experiments and the replications.

In 2011, Tay et al. reported *PTEN* was modulated through competing endogenous RNAs (ceRNAs), which are protein-coding RNA transcripts that compete for microRNAs through common micoRNA response elements (MREs). Testing four candidate *PTEN* ceRNAs Tay and colleagues reported that for three of these candidate ceRNAS (*SERINC1*, *VAPA*, and *CNOT6L*) silencing the ceRNAs impacted the activity of a luciferase construct engineered with the 3'UTR of *PTEN*

(*Tay et al., 2011*). Using the same reporter construct, overexpression of the 3'UTRs of these ceRNAs were reported to increase luciferase activity suggesting inhibition of *PTEN* 3'UTR was relieved (*Tay et al., 2011*). These effects were reported to be dependent on microRNAs since inhibition of PTEN protein expression when ceRNAs were depleted was abrogated when DICER, a key part of the microRNA machinery, was disrupted (*Tay et al., 2011*). Two ceRNAs, *VAPA* and *CNOT6L*, when depleted also resulted in increased cell proliferation and phosphorylation of AKT when depleted, which was attenuated when DICER was disrupted (*Tay et al., 2011*).

The Registered Report for the paper by *Tay et al., 2011* described the experiments to be replicated (Figures 3C-D, 3G-H, 5A-B, and Supplemental Figures S3A-B), and summarized the current evidence for these findings (*Phelps et al., 2016*). Since that publication additional studies have reported finding other ceRNAs of *PTEN*. *TNRC6B* was identified as a ceRNA of *PTEN* with depletion of *TNRC6B* reported to decrease PTEN mRNA and protein expression in the prostate cancer cell lines DU145, 22RV1, and BM1604 and increase cell proliferation in DU145 and PC3 cells (*Zarringhalam et al., 2017*). *Zarringhalam et al., 2017* also used *CNOT6L* in their study and reported depletion of *CNOT6L* produced similar results as *TNRC6B* depletion. *DNMT3B* and *TET3* were recently reported to be ceRNAs of *PTEN* with miR-4465 identified as a microRNA regulating these three transcripts via their 3'UTRs (*Roquid et al., 2019*). Multiple studies have reported a growing list of potential ceRNAs, which includes mRNAs, pseudogenes, lncRNAs, and circRNAs (*Gebert and MacRae, 2019*; *Li et al., 2018*; *Wang et al., 2016*; *Yang et al., 2016*). For example, the lncRNA *BGL3* has been identified as a ceRNA for *PTEN* to regulate Bcr-Abl-mediated cellular transformation in chronic myeloid leukemia (*Guo et al., 2015*) and c-Myc has been reported as a potential ceRNA for PML/RARα in acute promyelocytic leukemia (*Ding et al., 2016*). Prediction of putative ceRNAs are being reported using a variety of computational methods and data sources that construct ceRNA interaction networks (*Chen et al., 2018*; *Chiu et al., 2017*; *Feng et al., 2019*; *Liu et al., 2019*; *Park et al., 2018b*; *Song et al., 2016*; *Sun et al., 2016*; *Swain and Mallick, 2018*; *Wang et al., 2019*; *Yue et al., 2019*). Additionally, recently it has been reported that 3'UTR shortening of transcripts of predicted ceRNAs could be a potential mechanism of repressing tumor-suppressor genes, including *PTEN*, in trans by disrupting ceRNA cross-talk (*Park et al., 2018a*).

At the same time, whether physiological changes of ceRNAs can modulate microRNA activities remains controversial (*Cai and Wan, 2018*; *Thomson and Dinger, 2016*). Experiments designed to test the feasibility of the ceRNA hypothesis, have reported that microRNA-binding sites are generally much higher than the number of microRNA molecules (*Denzler et al., 2014*). This would suggest that under physiological conditions ceRNA perturbation would likely lead to a change too small to be detected and to produce biological consequences (*Broderick and Zamore, 2014*; *Denzler et al., 2014*). Mullokandov and colleagues reported that only the most abundant microRNAs mediate target suppression as over 60% of detected microRNAs have no discernable activity (*Mullokandov et al., 2012*). Bosson and colleagues suggested that the microRNA-target ratios determined the respective susceptibility to ceRNA-mediated regulation (*Bosson et al., 2014*). This model has been further examined and Denzler and colleagues reported that while microRNA levels did not affect site competition, they defined microRNA-mediated repression (*Denzler et al., 2016*). The experimental strategies currently used for studying the ceRNA hypothesis are also limited, especially when attempting to represent the in vivo levels of endogenous RNAs (*Cai and Wan, 2018*; *Jens and Rajewsky, 2015*; *Thomson and Dinger, 2016*).

The outcome measures reported in this Replication Study will be aggregated with those from the other Replication Studies to create a dataset that will be examined to provide evidence about reproducibility of cancer biology research, and to identify factors that influence reproducibility more generally.

## Results and discussion

### ceRNA depletion on *PTEN*-3'UTR luciferase reporter activity

We independently replicated an experiment to test if putative *PTEN* ceRNAs modulate the 3'UTR of *PTEN*. This experiment used a chimeric luciferase construct tagged with the *PTEN* 3'UTR (Luc-*PTEN*-

3'UTR) to uncouple regulation of *PTEN* via 3'UTR-targeting microRNAs from *PTEN* mRNA transcription and protein stability. This is similar to what was reported in Figure 3C and Supplemental Figure S3A of *Tay et al., 2011* and described in Protocol 1 in the Registered Report (*Phelps et al., 2016*). DU145 cells were co-transfected with Luc-*PTEN*-3'UTR and siRNAs targeting the same putative *PTEN* ceRNAs as the original study. Knockdown efficiency was examined by reverse transcription-quantitative polymerase chain reaction (RT-qPCR). The average reduction in gene expression relative to control siRNA was 65% when *SERINC1*, *VAPA*, *CNOT6L*, or *PTEN* were targeted, but was only 21% for *ZNF460* (*Figure 1—figure supplement 1C*), despite transfection efficiency being at least 90% based on a fluorescent transfection indicator (*Figure 1—figure supplement 1A*). Luciferase activity was decreased in *PTEN* depleted cells (average RLU = 12%) relative to control siRNA (average RLU = 100%); however, luciferase activity when the putative *PTEN* ceRNAs were targeted for depletion were similar to control siRNA (*Figure 1*, *Figure 1—figure supplement 1B*). All planned comparisons were not statistically significant (see *Figure 1* legend). The original study reported statistically significant decreased luciferase activity with siRNA-mediated depletion of *SERINC1* (average RLU = 70%), *VAPA* (average RLU = 48%), *CNOT6L* (average RLU = 70%), or *PTEN* (average RLU = 20%), but not for knockdown of *ZNF460* (average RLU = 109%), compared to control siRNA (average RLU = 100%) (*Tay et al., 2011*). The range of luciferase values reported in the original study had relative standard deviations (RSDs) (control = 9%; *SERINC1* = 10%; *VAPA* = 6%; *CNOT6L* = 5%; *ZNF460* = 8%; *PTEN* = 5%) that were much smaller than the RSDs observed in this replication attempt (control = 36%; *SERINC1* = 57%; *VAPA* = 36%; *CNOT6L* = 50%; *ZNF460* = 33%; *PTEN* = 19%), which is one of the factors that could influence if statistical significance is reached, particularly since the sample size of this replication attempt was determined a priori to detect the effect based on the originally reported data. The original study also reported an achieved knockdown of 90% or greater when *SERINC1*, *VAPA*, *CNOT6L*, or *PTEN* were targeted, but was 65% for *ZNF460* (*Tay et al., 2011*). The difference in achieved knockdown between the original study and this replication attempt is a possible reason for the differences in Luc-*PTEN*-3'UTR outcomes. A higher level of knockdown might be required to observe an effect with this experimental design. Although, unlike experiments that evaluate protein function where a higher level of knockdown or a longer period of time is usually needed to observe a phenotype (*Curtis and Nardulli, 2009*; *O'Keefe, 2013*), the putative ceRNA function of these mRNAs should correspond to the level of knockdown. Thus, a 65% knockdown would have been expected to capture ~72% of the effect observed in the original study that reported a 90% knockdown. To summarize, for this experiment, we found results that were not statistically significant where predicted, varied in direction relative to the original study for the putative *PTEN* ceRNAs, and in the same direction as the original study for cells transfected with siPTEN.

## ceRNA overexpression on *PTEN*-3'UTR luciferase reporter activity

To test if sequestration of the putative *PTEN* ceRNAs impacted *PTEN* expression, we ectopically overexpressed the 3'UTR of the same putative *PTEN* ceRNAs as the original study in DU145 cells along with the Luc-*PTEN*-3'UTR plasmid. This is similar to what was reported in Figure 3D of *Tay et al., 2011* and described in Protocol 2 in the Registered Report (*Phelps et al., 2016*). We used the same plasmids as the original study, which cloned the 3'UTRs of *VAPA* and *CNOT6L* as two separate fragments due to their large size with the fragments subdivided based on location of predicted MREs (*Tay et al., 2011*). We found that compared to cells transfected with empty vector control, cells transfected with 3'UTR of the putative *PTEN* ceRNA plasmids or the 3'UTR of *PTEN* had decreased luciferase activity (*Figure 2*, *Figure 2—figure supplement 1*). The planned comparisons were statistically significant for *SERINC1* 3'UTR, *VAPA* 3'UTR2, *CNOT6L* 3'UTR1, *CNOT6L* 3'UTR2, and *PTEN* 3'UTR1, but not for *VAPA* 3'UTR1 (see *Figure 2* legend). The original study reported statistically significant increased luciferase activity with *SERINC1* 3'UTR (average RLU = 128%), *VAPA* 3'UTR1 (average RLU = 141%), *VAPA* 3'UTR2 (average RLU = 150%), *CNOT6L* 3'UTR1 (average RLU = 143%), *CNOT6L* 3'UTR2 (average RLU = 146%), or *PTEN* 3'UTR (average RLU = 153%) compared to empty vector control (average RLU = 100%) (*Tay et al., 2011*). The range of luciferase values reported in the original study had RSDs (control = 9%; *SERINC1* 3'UTR = 9%; *VAPA* 3'UTR1 = 13%; *VAPA* 3'UTR2 = 6%; *CNOT6L* 3'UTR1 = 7%; *CNOT6L* 3'UTR2 = 7%; *PTEN* 3'UTR = 1%) that were smaller than the RSDs observed in this replication attempt (control = 17%; *SERINC1* 3'UTR = 12%; *VAPA* 3'UTR1 = 12%; *VAPA* 3'UTR2 = 15%; *CNOT6L* 3'UTR1 = 15%;

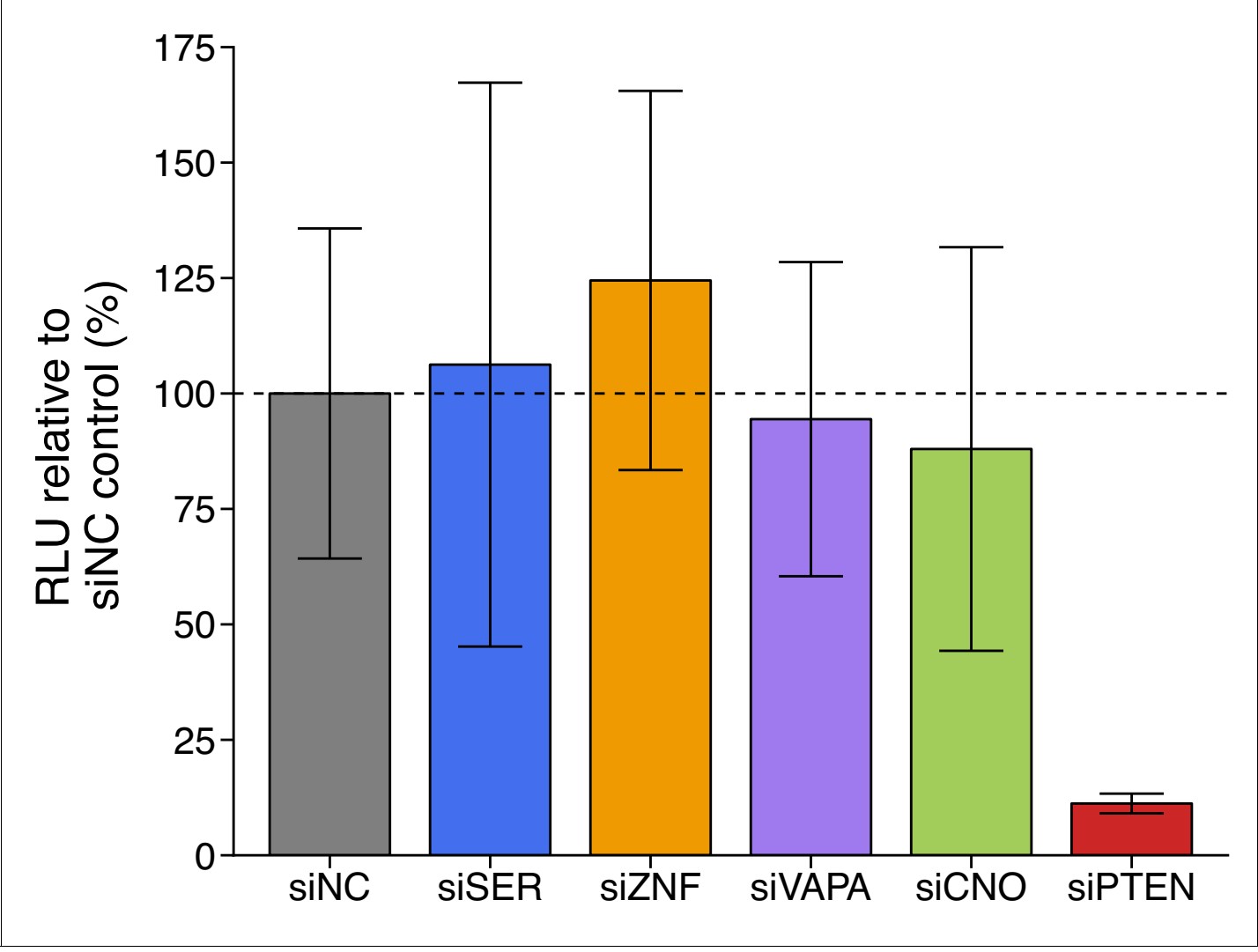

**Figure 1.** Luciferase activity in DU145 cells co-transfected with siRNA against *PTEN* ceRNAs and a luciferase-*PTEN* 3'UTR reporter construct. DU145 cells were transfected with a luciferase reporter with a fragment of the 3'UTR of *PTEN*. Cells were also co-transfected with non-targeting control siRNA (siNC) or siRNA plasmids targeting *SERINC1* (siSER), *ZNF460* (siZNF), *VAPA* (siVAPA), *CNOT6L* (siCNO), or *PTEN* (siPTEN). Cells were harvested 72 hr later for luciferase activity. Relative luminescence unit (RLU) is presented for each condition relative to the siNC condition. Means reported and error bars represent *SD* from four independent biological repeats. Two-sample *t*-test of RLU values between siNC and siSER: $t(6) = 0.177$, uncorrected $p=0.866$ with a priori Bonferroni adjusted significance threshold of 0.01, Bonferroni corrected $p>0.99$; siNC and siZNF: $t(6) = 0.899$, uncorrected $p=0.403$, Bonferroni corrected $p>0.99$; siNC and siVAPA: $t(6) = 0.225$, uncorrected $p=0.829$, Bonferroni corrected $p>0.99$; siNC and siCNO: $t(6) = 0.426$, uncorrected $p=0.685$, Bonferroni corrected $p>0.99$; Wilcoxon-Mann-Whitney test of RLU values between siNC and siPTEN: $U = 16$, uncorrected $p=0.029$, Bonferroni corrected $p=0.143$. Additional details for this experiment can be found at https://osf.io/spv4f/.
The online version of this article includes the following figure supplement(s) for figure 1:

**Figure supplement 1.** Knockdown efficiency and individual repeats of luciferase-*PTEN* 3'UTR reporter assay in DU145 cells co-transfected with siRNA against *PTEN* ceRNAs.

*CNOT6L* 3'UTR2 = 14%; *PTEN* 3'UTR = 10%). To summarize, we found results that were statistically significant (with the exception of *VAPA* 3'UTR1) and in the opposite direction as the original study.

## ceRNA depletion on PTEN expression

We replicated an experiment to test the microRNA dependency of the putative *PTEN* ceRNAs. This experiment used the same isogenic wild-type and DICER mutant (Dicer[Ex5]) HCT116 colon carcinoma cells as the original study. The Dicer[Ex5] cell line, which was engineered to disrupt a well-conserved segment of the N-terminal helicase domain in exon 5 of DICER, while leaving the RNase III domains

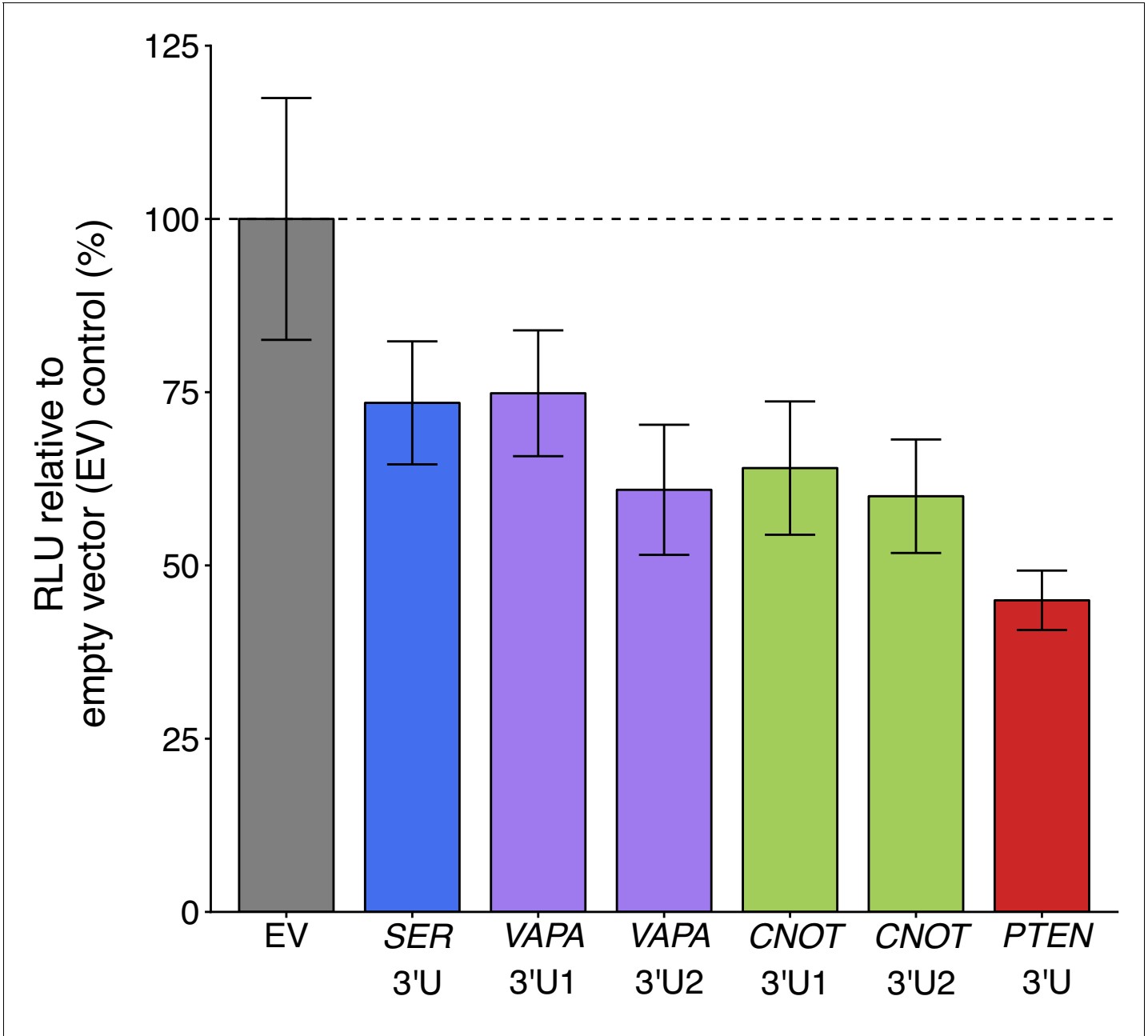

**Figure 2.** Luciferase activity in DU145 cells co-transfected with 3'UTR of *PTEN* ceRNAs and a luciferase-*PTEN* 3'UTR reporter construct. DU145 cells were transfected with a luciferase reporter with a fragment of the 3'UTR of *PTEN*. Cells were also co-transfected with empty vector (EV) or plasmids that express the 3'UTR of *SERINC1* (*SER* 3'U), *VAPA* (*VAPA* 3'U1 and *VAPA* 3'U2), *CNOT6L* (*CNOT* 3'U1 and *CNOT* 3'U2), or *PTEN* (*PTEN* 3'U). Cells were harvested 72 hr later for luciferase activity. Relative luminescence unit (RLU) is presented for each condition relative to the EV condition. Means reported and error bars represent *SD* from six independent biological repeats. Two-sample *t*-test of RLU values between *SER* 3'U and EV: $t(10) = 3.32$, uncorrected $p=0.0077$ with a priori Bonferroni adjusted significance threshold of 0.0083, Bonferroni corrected $p=0.046$; *VAPA* 3'U1 and EV: $t(10) = 3.13$, uncorrected $p=0.011$, Bonferroni corrected $p=0.064$; *VAPA* 3'U2 and EV: $t(10) = 4.83$, uncorrected $p=6.90 \times 10^{-4}$, Bonferroni corrected $p=0.0041$; *CNOT* 3'U1 and EV: $t(10) = 4.42$, uncorrected $p=0.0013$, Bonferroni corrected $p=0.0078$; *CNOT* 3'U2 and EV: $t(7.1) = 5.09$, uncorrected $p=0.0014$, Bonferroni corrected $p=0.0082$; *PTEN* 3'U and EV: $t(5.6) = 7.50$, uncorrected $p=3.99 \times 10^{-4}$, Bonferroni corrected $p=0.0024$. Additional details for this experiment can be found at https://osf.io/mryvq/.

The online version of this article includes the following figure supplement(s) for figure 2:

**Figure supplement 1.** Individual repeats of luciferase-*PTEN* 3'UTR reporter assay in DU145 cells co-transfected with 3'UTR of *PTEN* ceRNAs.

intact, displays a hypomorphic phenotype in the processing of mature microRNAs (*Cummins et al., 2006*). This experiment is similar to what was reported in Figure 3G–H and Supplemental Figure S3B of *Tay et al., 2011* and described in Protocol 3 in the Registered Report (*Phelps et al., 2016*). Wild-type and Dicer[Ex5] HCT116 cells were transfected with siRNAs targeting the same putative *PTEN* ceRNAs as the original study. Knockdown efficiency, measured by RT-qPCR, revealed the average reduction in gene expression relative to control siRNA was 81% in both cell lines for all putative *PTEN* ceRNAs, with the greatest biological variability in Dicer[Ex5] HCT116 cells when targeting *CNOT6L* (*Figure 3—figure supplement 1B*).

Depletion of the putative *PTEN* ceRNAs resulted in downregulation of PTEN protein in wild-type HCT116 cells to an average of 80%, 43%, or 61% for siRNA-mediated depletion of *SERINC1*, *VAPA*, or *CNOT6L*, respectively, relative to control siRNA (average PTEN expression = 100%) (*Figure 3A–B*, *Figure 3—figure supplement 1A*). As a control, siRNAs targeting *PTEN* reduced PTEN protein levels to an average of 1.6%. To compare the relative PTEN expression among the various conditions, we planned to conduct four comparisons using the Bonferroni correction to adjust for multiple comparisons. The comparison of PTEN protein levels between control siRNA and siRNA targeting *VAPA*, *CNOT6L*, or *PTEN* were statistically significant, while the comparison of control siRNA and siRNA targeting *SERINC1* were not (see *Figure 3* legend). The original study reported statistically significant decreases in PTEN protein levels with siRNA-mediated depletion of *SERINC1* (average PTEN expression = 53%), *VAPA* (average PTEN expression = 52%), *CNOT6L* (average PTEN expression = 59%), or *PTEN* (average PTEN expression = 1.9%) compared to control siRNA (average PTEN expression = 100%) in wild-type HCT116 cells (*Tay et al., 2011*).

For Dicer[Ex5] HCT116 cells, we found depletion of the putative *PTEN* ceRNAs resulted in higher PTEN protein levels (*SERINC1*: 290%; *VAPA*: 144%; *CNOT6L*: 256%) relative to control siRNA (average PTEN expression = 100%), while targeting *PTEN* reduced PTEN protein levels to an average of 2.6%. (*Figure 3A–B*, *Figure 3—figure supplement 1A*). To compare the relative PTEN expression among the various conditions, a similar analysis as described above for wild-type HCT116 cells was performed for Dicer[Ex5] HCT116 cells. We found that PTEN protein levels between control siRNA and siRNA targeting *SERINC1* or *PTEN* were statistically significant, while the comparisons between control siRNA and siRNA targeting *VAPA* or *CNOT6L* were not (see *Figure 3* legend). The original study reported PTEN downregulation by ceRNA depletion was attenuated in Dicer[Ex5] HCT116 cells with average PTEN expression around the same as control siRNA (control siRNA: 100%; *SERINC1*: 117%; *VAPA*: 108%; *CNOT6L*: 113%), while the average PTEN expression in cells transfected with siRNA-mediated depletion of *PTEN* was 1.3% (*Tay et al., 2011*). Similar to the siRNA-mediated depletion of putative *PTEN* ceRNA in DU145 cells described above, the original study reported a knockdown of greater than 90% for most conditions (*Tay et al., 2011*). The level of knockdown required to yield a given phenotype varies because it is system-dependent (*Bailoo et al., 2014*), thus the difference in achieved knockdown between the original study and this replication attempt should be considered when interpreting these results. Further, the original study reported lower RSDs for PTEN protein levels across all the siRNA conditions in the Dicer[Ex5] HCT116 cells compared to the wild-type HCT116 cells (Dicer[Ex5]: 0.1–9% vs wild-type: 8–17%), while this replication attempt observed larger RSDs compared to the original study, especially for Dicer[Ex5] HCT116 cells (Dicer[Ex5]: 29–61% vs wild-type: 12–46%). Importantly, the individual biological repeats were largely consistent relative to the control siRNA condition (*Figure 3—figure supplement 1A*). This difference in variance between the original study and this replication attempt could influence if statistical significance is reached. To summarize, for this experiment, we found results that were generally in the same direction as the original study, varied in terms of statistical significance, and in Dicer[Ex5] HCT116 cells effects that were of a larger magnitude than the original study. This absence of an attenuated ceRNA effect in this replication attempt suggests the null hypothesis that there is no difference in PTEN protein expression when the microRNA machinery is disrupted can be rejected.

## ceRNA depletion on cell proliferation

We replicated an experiment to evaluate cell proliferation of DU145, wild-type HCT116, and Dicer[Ex5] HCT116 cells in response to siRNA-mediated silencing of the putative *PTEN* ceRNAs. This is similar to what was reported in Figure 5B of *Tay et al., 2011* and described in Protocol 4 in the Registered Report (*Phelps et al., 2016*). Cells were transfected with siRNAs targeting the same putative *PTEN* ceRNAs as the original study. Knockdown efficiency, measured by RT-qPCR, revealed an average

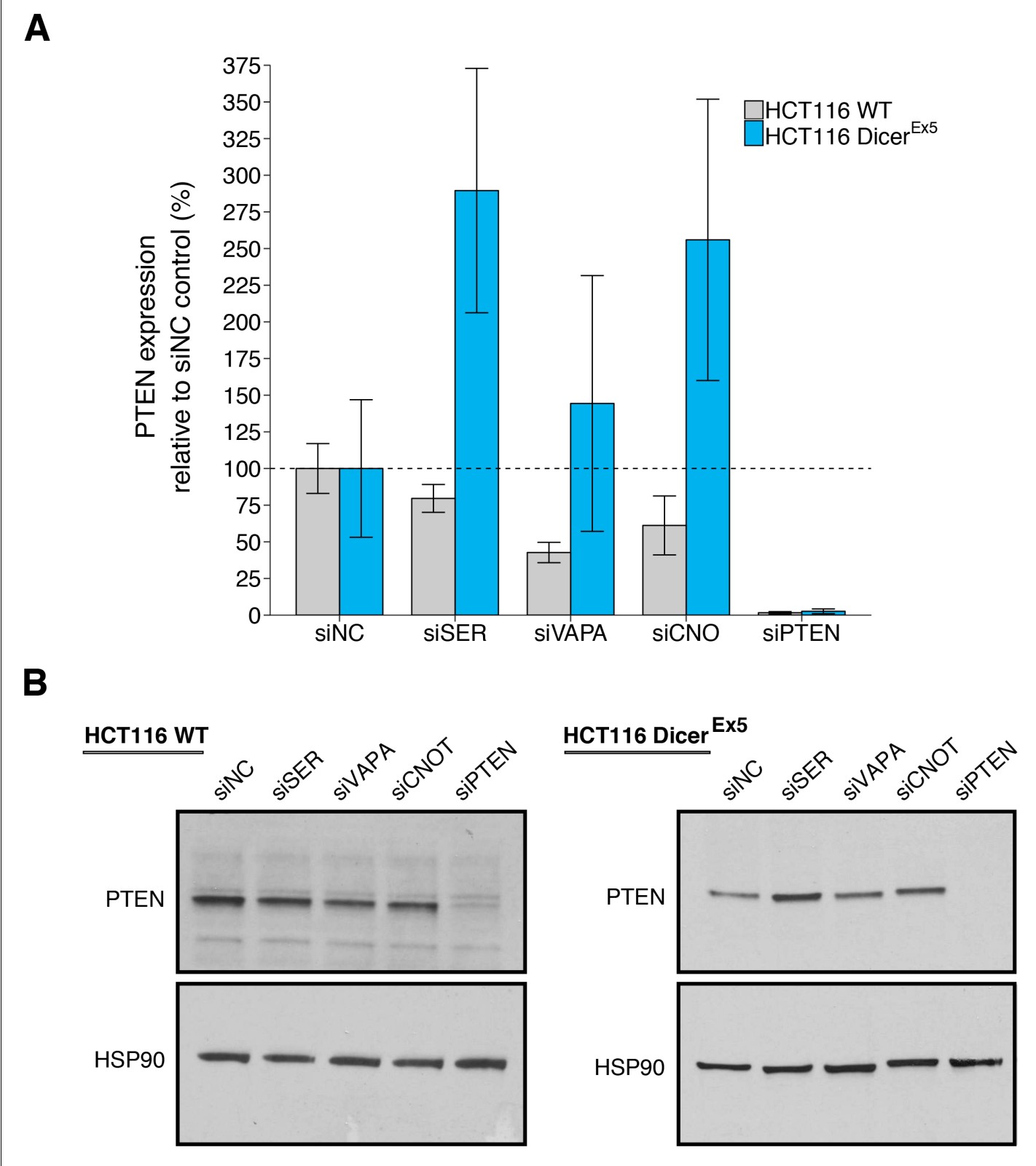

**Figure 3.** PTEN protein expression in wild-type and DICER mutant HCT116 cells depleted of *PTEN* ceRNAs. Wild-type (WT) and DICER mutant (Dicer^Ex5) HCT116 cells were transfected with non-targeting control siRNA (siNC) or siRNA plasmids targeting *SERINC1* (siSER), *VAPA* (siVAPA), *CNOT6L* (siCNO), or *PTEN* (siPTEN). Cells were harvested 72 hr later for Western blot analysis. (**A**) Relative protein expression (PTEN/HSP90) are presented for each condition. Western blot bands were quantified, PTEN levels were normalized to HSP90, with protein expression presented relative
*Figure 3 continued on next page*

Figure 3 continued

to siNC. Means reported and error bars represent *SD* from three independent biological repeats for wild-type HCT116 cells and four repeats for Dicer$^{Ex5}$ HCT116 cells. Analysis of wild-type HCT116 cells: one-way ANOVA (equal variance) on PTEN/HSP90 expression: $F(4,10) = 25.4$, I$=3.18\times10^{-5}$. Planned contrasts between siNC and siSER: $t(10) = 1.94$, uncorrected I$=0.082$ with a priori Bonferroni adjusted significance threshold of 0.0125, Bonferroni corrected $p=0.326$; siNC and siVAPA: $t(10) = 5.44$, uncorrected $p=2.85\times10^{-4}$, Bonferroni corrected $p=0.0011$; siNC and siCNOT: $t(10) = 3.69$, uncorrected $p=0.0042$, Bonferroni corrected $p=0.017$; siNC and siPTEN: $t(10) = 9.34$, uncorrected $p=2.97\times10^{-6}$, Bonferroni corrected $p=1.19\times10^{-5}$. Analysis of Dicer$^{Ex5}$ HCT116 cells: one-way ANOVA (unequal variance) on PTEN/HSP90 expression: $F(4,6.0) = 19.3$, $p=0.0014$. Planned comparisons: siNC and siSER: two-sample *t*-test, $t(6) = 3.96$, uncorrected $p=0.0074$ with a priori Bonferroni adjusted significance threshold of 0.0125, Bonferroni corrected $p=0.030$; siNC and siVAPA: two-sample *t*-test, $t(6) = 0.896$, uncorrected $p=0.405$, Bonferroni corrected $p>0.99$; siNC and siCNOT: Welch's *t*-test, $t(4.36) = 2.92$, uncorrected $p=0.039$, Bonferroni corrected $p=0.156$; siNC and siPTEN: two-sample *t*-test, $t(6) = 4.15$, uncorrected $p=0.0060$, Bonferroni corrected $p=0.024$. (B) Representative Western blots probed with an anti-PTEN antibody and anti-HSP90 antibody. Additional details for this experiment can be found at https://osf.io/drcbw/.

The online version of this article includes the following figure supplement(s) for figure 3:

**Figure supplement 1.** Knockdown efficiency and individual repeats of PTEN protein expression in wild-type and DICER mutant HCT116 cells transfected with siRNA against *PTEN* ceRNAs.

reduction in gene expression relative to control siRNA was 79% when considering all cell lines (***Figure 4—figure supplement 1B***). Proliferation activity was determined using the crystal violet assay starting the day after transfection with results presented as the difference in the values at the start of the timecourse for each condition (i.e. for each condition the value at the start of the timecourse was set to 0), similar to the original study. For DU145 cells, we found that siRNA-mediated depletion of *VAPA* or *PTEN* resulted in increased cell proliferation compared to cells transfected with control siRNA, while depletion of *CNOT6L* resulted in decreased cell proliferation (***Figure 4***, ***Figure 4—figure supplement 1A***). The area under the curve (AUC) during the timecourse for each biological repeat was used to compare each condition to the control siRNA, which were not statistically significant (see ***Figure 4*** legend). The original study reported siRNA-mediated targeting of *VAPA*, *CNOT6L*, or *PTEN* in DU145 cells resulted in a statistically significant increase in proliferation compared to control siRNA (***Tay et al., 2011***). The range of AUC values reported in the original study had a RSD for the control condition (26%) similar to this replication study (20%); however, the RSDs for the other conditions were much lower in the original study (*VAPA* = 4%; *CNOT6L* = 9%; *PTEN* = 7%) then this replication attempt (*VAPA* = 26%; *CNOT6L* = 20%; *PTEN* = 10%). As stated above this difference in variance between the original study and this replication attempt is a factor that could influence if statistical significance is reached.

For wild-type HCT116 cells, we found that compared to cells transfected with control siRNA, depletion of *VAPA*, *CNOT6L*, or *PTEN* resulted in different levels of increased cell proliferation (***Figure 4***, ***Figure 4—figure supplement 1A***). For Dicer$^{Ex5}$ HCT116 cells, depletion of *PTEN* resulted in increased cell proliferation compared to control siRNA with a similar magnitude as wild-type cells, depletion of *VAPA* resulted in an increased proliferation compared to control siRNA, but not at the same magnitude as occurred in wild-type cells, while depletion of *CNOT6L* resulted in a slight decrease in cell proliferation compared to control siRNA. To test if depletion of the putative *PTEN* ceRNAs increased proliferation in wild-type HCT116 cells and were attenuated in the Dicer$^{Ex5}$ HCT116 cells, we performed an analysis of variance (ANOVA) on the AUC for each biological repeat. The ANOVA result was statistically significant for the siRNA main effect ($F(3,24) = 12.1$, $p=5.20\times10^{-5}$). Thus, the null hypothesis that there is no difference in cell proliferation when the putative *PTEN* ceRNAs or *PTEN* was depleted, whether or not it was conducted in wild-type or Dicer$^{Ex5}$ HCT116 cells, can be rejected. The main effect for cell type ($F(1,24) = 1.81$, $p=0.191$) was not statistically significant, indicating the null hypothesis that there is no difference in cell proliferation between wild-type or Dicer$^{Ex5}$ HCT116 cells can not be rejected, and the interaction effect was not statistically significant ($F(3,24) = 0.734$, $p=0.542$). These results suggest that while there were differences in cell proliferation when the putative *PTEN* ceRNAs or *PTEN* were depleted, it was similar between cell lines suggesting a lack of an attenuated ceRNA effect. We also conducted six comparisons using the Bonferroni correction to adjust for multiple comparisons, making the a priori adjusted significance threshold 0.0083. According to this criterion, depletion of *PTEN* in wild-type or Dicer$^{Ex5}$ HCT116 cells resulted in statistically significant increases in cell proliferation compared to control siRNA. Depletion of *VAPA* or *CNOT6L* did not result in a statistically significant increase in cell

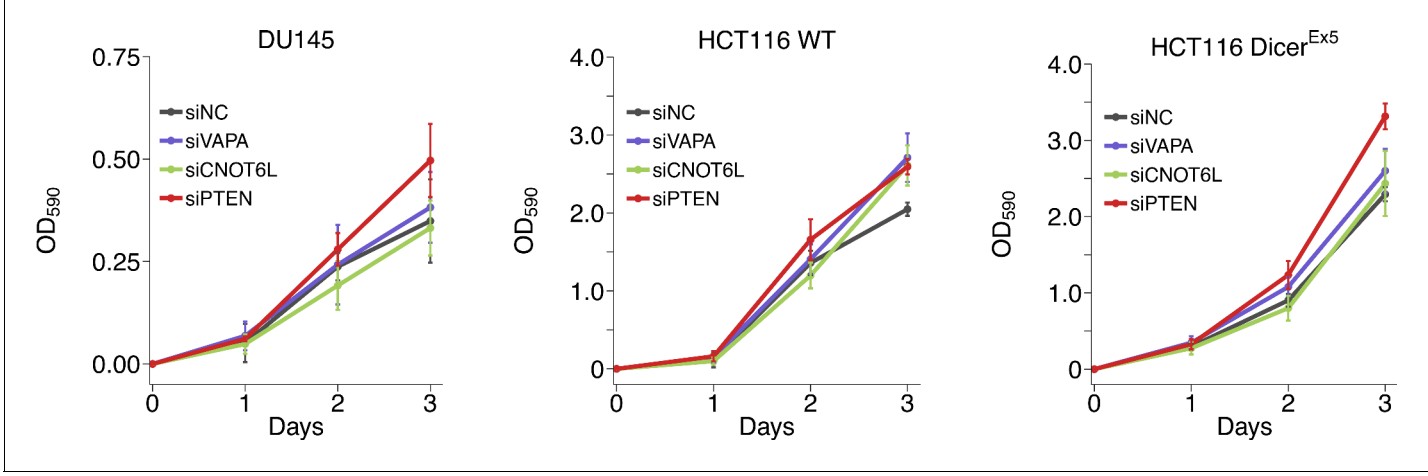

**Figure 4.** Growth of cells depleted of *PTEN* ceRNAs. DU145, wild-type (WT) and DICER mutant (Dicer[Ex5]) HCT116 cells were transfected with either a non-targeting control siRNA (siNC) or siRNA plasmids targeting *VAPA* (siVAPA), *CNOT6L* (siCNO), or *PTEN* (siPTEN). Crystal violet proliferation assays were performed each day as indicated starting the day after transfection. Relative $OD_{590}$ was calculated relative to the average Day 0 values for each condition. Means reported and error bars represent *SD* from five independent biological repeats for DU145 cells and four times for HCT116 WT and Dicer[Ex5] cells. Analysis on the area under the curve (AUC) for each condition of each biological repeat (reported as dot plot in ***Figure 4—figure supplement 1A***). Analysis results for DU145 cells: one-way ANOVA (equal variance): $F_{(3,16)} = 3.27$, $p=0.049$. Planned contrasts between siNC and siVAPA: $t_{(16)} = 0.648$, uncorrected $p=0.526$ with a priori Bonferroni adjusted significance threshold of 0.0167, Bonferroni corrected $p>0.99$; siNC and siCNOT6L: $t_{(16)} = 0.950$, uncorrected $p=0.356$, Bonferroni corrected $p>0.99$; siNC and siPTEN: $t_{(16)} = 2.09$, uncorrected $p=0.053$, Bonferroni corrected $p=0.158$. Analysis of HCT116 cells: two-way ANOVA interaction between DICER status (wild-type or Ex5) and siRNA target: $F_{(3,24)} = 0.734$, $p=0.542$; main effect of DICER status: $F_{(1,24)} = 1.81$, $p=0.191$; main effect of siRNA target: $F_{(3,24)} = 12.1$, $p=5.20\times10^{-5}$. Planned contrasts in HCT116 WT cells: siNC and siVAPA: $t_{(24)} = 2.02$, uncorrected $p=0.054$ with a priori Bonferroni adjusted significance threshold of 0.0083, Bonferroni corrected $p=0.325$; siNC and siCNOT6L: $t_{(24)} = 0.506$, uncorrected $p=0.618$, Bonferroni corrected $p>0.99$; siNC and siPTEN: $t_{(24)} = 3.03$, uncorrected $p=0.0057$, Bonferroni corrected $p=0.034$. Planned contrasts in HCT116 DICER[Ex5] cells: siPTEN and siVAPA: $t_{(24)} = 2.43$, uncorrected $p=0.023$, Bonferroni corrected $p=0.138$; siPTEN and siCNOT6L: $t_{(24)} = 4.57$, uncorrected $p=1.25\times10^{-4}$, Bonferroni corrected $p=7.48\times10^{-4}$; siNC and siPTEN: $t_{(24)} = 4.31$, uncorrected $p=2.42\times10^{-4}$, Bonferroni corrected $p=0.0015$. Additional details for this experiment can be found at https://osf.io/5c7sb/. The online version of this article includes the following figure supplement(s) for figure 4:

**Figure supplement 1.** Knockdown efficiency and individual repeats of cell growth assay in cells transfected with siRNA against *PTEN* ceRNAs.

---

proliferation compared to control siRNA in wild-type HCT116 cells. Additionally, depletion of *CNOT6L*, but not *VAPA*, resulted in a statistically significant decrease in cell proliferation compared to *PTEN*-depleted Dicer[Ex5] HCT116 cells. The original study reported reduced expression of *VAPA* or *CNOT6L* in wild-type HCT116 cells resulted in a statistically significant increase in cell proliferation compared to control siRNA similar to what was observed with *PTEN* siRNA, which was statistically significantly attenuated in the Dicer[Ex5] HCT116 cells (***Tay et al., 2011***). Further, the original study (Dicer[Ex5]: 1–15%; wild-type: 5–10%) and this replication attempt (Dicer[Ex5]: 6–19%; wild-type: 5–10%) observed similar RSDs. To summarize, for this experiment we found results that varied in statistical significance and varied in direction relative to the original study for the putative *PTEN* ceRNAs, but were in the same direction as the original study for cells transfected with siPTEN.

## Meta-analyses of original and replication effects

We performed a meta-analysis using a random-effects model, where possible, to combine each of the effects described above as pre-specified in the confirmatory analysis plan (***Phelps et al., 2016***). To provide a standardized measure of the effect, a common effect size was calculated for each effect from the original and replication studies. Cohen's *d* is the standardized difference between two means using the pooled sample standard deviation, while the effect size Glass' delta is the standardized difference between two means using the standard deviation of only the control group. Glass' delta was used when the variance between the control and treatment conditions were not equal in the original or replication study experiments. The estimate of the effect size of one study, as well as the associated uncertainty (i.e. confidence interval), compared to the effect size of the other study

provides one approach to compare the original and replication results (*Errington et al., 2014*; *Valentine et al., 2011*). Importantly, the width of the confidence interval (CI) for each study is a reflection of not only the confidence level (e.g. 95%), but also variability of the sample (e.g. *SD*) and sample size.

There were five comparisons of the *PTEN*-3'UTR luciferase reporter activity when putative *PTEN* ceRNAs were depleted, which were reported in *Figure 1* of this study and Figure 3C of *Tay et al., 2011*. Only one of the effects, control siRNA compared to *ZNF460* siRNA, was consistent in direction and when considering if the effect size point estimate of each study was within the confidence interval of the other study, suggesting the null hypothesis that there is no difference in reporter activity can not be rejected (*Figure 5A*). The other effects were inconsistent in whether the direction of the effect was the same between the two studies, if the effect size of one study was within the confidence interval of the other study, or both. Additionally, the meta-analyses were not statistically significant, with all but one of the effects having large confidence intervals around the meta-analysis effect size along with statistically significant Cochran's *Q* tests (siNC and siSERINC1, p=0.013; siNC and siVAPA, p=0.0019; siNC and siPTEN, p=0.0054) suggesting heterogeneity between the original and replication studies.

*PTEN*-3'UTR luciferase reporter activity was also tested when the 3'UTR of putative *PTEN* ceRNAs were ectopically overexpressed, reported in *Figure 2* of this study and Figure 3D of *Tay et al., 2011*. The direction of all six comparisons were in the opposite direction of the original study with none of the effect size point estimates within the confidence intervals of the other study (*Figure 5B*). The meta-analyses were not statistically significant, with all effects having large confidence intervals around the meta-analysis effect size along with statistically significant Cochran's *Q* tests (*SERINC1* 3'UTR and empty vector, p=0.0011; *VAPA* 3'UTR1 and empty vector, p=$2.55\times10^{-4}$; *VAPA* 3'UTR2 and empty vector, p=$1.28\times10^{-6}$; *CNOT6L* 3'UTR1 and empty vector, p=$1.07\times10^{-5}$; *CNOT6L* 3'UTR2 and empty vector, p=$2.54\times10^{-5}$; *PTEN* 3'UTR and empty vector, p=$1.86\times10^{-6}$) suggesting heterogeneity between the original and replication studies.

PTEN protein expression was examined in two cell lines, wild-type and Dicer[Ex5] HCT116 cells, following depletion of putative *PTEN* ceRNAs with four comparisons made in each cell line, which were reported in *Figure 3* of this study and Figure 3G–H of *Tay et al., 2011*. In wild-type cells, all effects were consistent when considering the direction of the effect and varied in whether the studies were within the confidence interval of the other study (*Figure 5C*). The meta-analysis of one of the effects, control siRNA compared to *SERINC1* siRNA, was not statistically significant suggesting the null hypothesis that there is no difference in PTEN protein expression can not be rejected; however, the large confidence intervals along with a statistically significant Cochran's *Q* tests (*p*=0.024) suggests heterogeneity between the original and replication studies. The meta-analyses of the other effects were statistically significant suggesting the null hypothesis can be rejected and that these ceRNAs regulate PTEN protein expression in HCT116 cells. In Dicer[Ex5] cells all effects were consistent when considering the direction of the effect and three had effect size point estimates of the original study that was within the confidence interval of the replication and vice versa. The meta-analyses of control siRNA compared to *SERINC1* or *CNOT6L* were statistically significant, suggesting the null hypothesis that there is no difference in PTEN protein expression when the microRNA machinery is disrupted can be rejected. The other two meta-analyses were not statistically significant suggesting the null hypothesis can not be rejected; however, for the control siRNA to *PTEN* siRNA comparison, the large confidence intervals along with a statistically significant Cochran's *Q* tests (*p*=$2.11\times10^{-9}$) suggests heterogeneity between the original and replication studies.

Cell proliferation was also tested when putative *PTEN* ceRNAs were depleted, with three comparisons made in each cell line as reported in *Figure 4* of this study and Figure 5B of *Tay et al., 2011*. In wild-type and Dicer[Ex5] HCT116 cells all effects were consistent when considering the direction of the effect and varied in whether the studies were within the confidence interval of the other study (*Figure 5D*). Additionally, the meta-analyses varied in terms of statistical significance with some meta-analyses having wide confidence intervals and statistically significant Cochran's *Q* tests (WT HCT116: siVAPA and siNC, *p*=0.048; siCNOT6L and siNC, *p*=$2.79\times10^{-4}$; siPTEN and siNC, *p*=0.0079; Dicer[Ex5] HCT116: siPTEN and siNC, *p*=0.029). In DU145 cells, the effects were inconsistent in whether the direction of the effect was the same between the two studies or if the effect size of one study was within the confidence interval of the other study. For all effects, the meta-analysis was not statistically significant, with wide confidence intervals, and statistically significant Cochran's

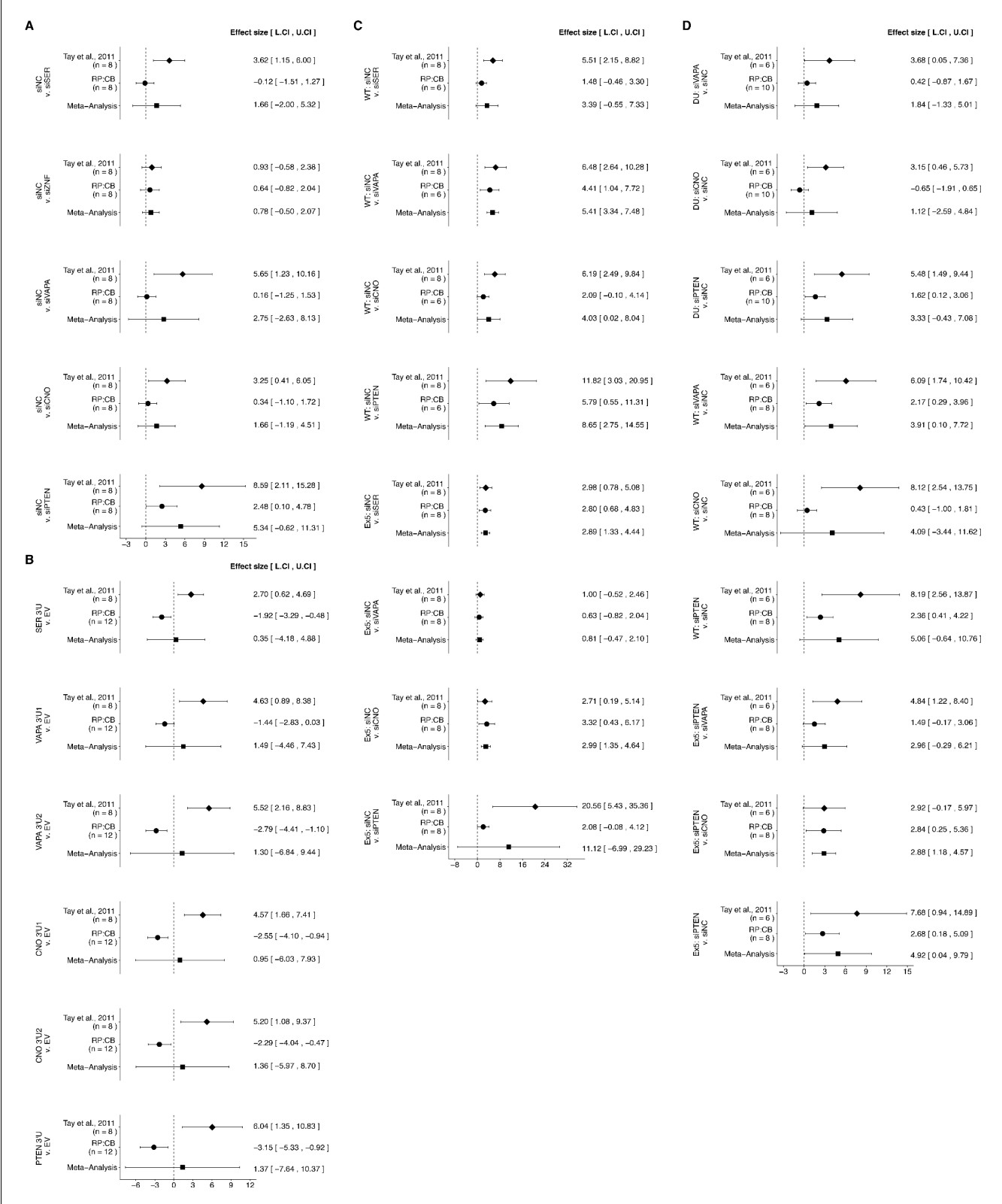

**Figure 5.** Meta-analyses of each effect. Effect size and 95% confidence interval are presented for *Tay et al., 2011*, this replication study (RP:CB), and a random effects meta-analysis of those two effects. For each effect, Cohen's *d* or Glass' delta, which are standardized differences between the two indicated measurements, is reported. Sample sizes used in *Tay et al., 2011* and RP:CB are reported under the study name. (A) These effects are related to the change in luciferase activity between the conditions reported in *Figure 1* of this study and Figure 3C of *Tay et al., 2011*. Meta-analysis *p*

*Figure 5 continued on next page*

*Figure 5 continued*

values: siNC and siSER (p=0.374); siNC and siZNF (p=0.233); siNC and siVAPA (p=0.316); siNC and siCNO (p=0.253); siNC and siPTEN (p=0.079). (**B**) These effects are related to the change in luciferase activity between the conditions reported in *Figure 2* of this study and Figure 3D of *Tay et al., 2011*. Meta-analysis *p* values: *SER* 3'U and EV (p=0.881); *VAPA* 3'U1 and EV (p=0.624); *VAPA* 3'U2 and EV (p=0.754); *CNO* 3'U1 and EV (p=0.790); *CNO* 3'U2 and EV (p=0.716); *PTEN* 3'U and EV (p=0.766). (**C**) These effects are related to the differences in PTEN protein expression between the conditions reported in *Figure 3* of this study and Figure 3H of *Tay et al., 2011*. Meta-analysis *p* values: WT HCT116: siNC and siSER (p=0.091); siNC and siVAPA (p=$2.99\times10^{-7}$); siNC and siCNO (p=0.049); siNC and siPTEN (p=0.0041): Dicer$^{Ex5}$ HCT116: siNC and siSER (p=$2.78\times10^{-4}$); siNC and siVAPA (p=0.215); siNC and siCNO (p=$3.70\times10^{-4}$); siNC and siPTEN (p=0.229). (**D**) These effects are related to the differences in cell growth between the conditions reported in *Figure 4* of this study and Figure 5B of *Tay et al., 2011*. Meta-analysis *p* values: DU145: siVAPA and siNC (p=0.255); siCNO and siNC (p=0.554); siPTEN and siNC (p=0.082): WT HCT116: siVAPA and siNC (p=0.045); siCNO and siNC (p=0.287); siPTEN and siNC (p=0.082): Dicer$^{Ex5}$ HCT116: siVAPA and siPTEN (p=0.075); siCNO and siPTEN (p=0.001); siPTEN and siNC (p=0.048). Additional details for these meta-analyses can be found at https://osf.io/xgrqp/.

*Q* tests (DU145: siVAPA and siNC, *p*=0.046; siCNOT6L and siNC, *p*=0.015, siPTEN and siNC, *p*=0.037), suggesting study heterogeneity.

This direct replication provides an opportunity to understand the present evidence of these effects. Any known differences, including reagents and protocol differences, were identified prior to conducting the experimental work and described in the Registered Report (*Phelps et al., 2016*). However, this is limited to what was obtainable from the original paper and through communication with the original authors, which means there might be particular features of the original experimental protocol that could be critical, but unidentified. So while some aspects, such as cell lines, antibodies, and the specific siRNA sequences and plasmids were maintained, others were unknown or not easily controlled for. These include variables such as cell line genetic drift (*Ben-David et al., 2018*; *Hughes et al., 2007*; *Kleensang et al., 2016*) and impacts of atmospheric oxygen on cell viability and growth (*Boregowda et al., 2012*). Whether these or other factors influence the outcomes of this study is open to hypothesizing and further investigation, which is facilitated by direct replications and transparent reporting.

# Materials and methods

## Key resources table

| Reagent type (species) or resource | Designation | Source or reference | Identifiers | Additional information |
|---|---|---|---|---|
| Cell line (*Homo sapiens*, male) | DU145 | ATCC | cat#:HTB-81; RRID:CVCL_0105 | |
| Cell line (*H. sapiens*, male) | Wild-type HCT116 cells | Horizon Discovery | cat# HD R02-019; RRID:CVCL_HD76 | |
| Cell line (*H. sapiens*, male) | DICER$^{Ex5}$ HCT116 cells | Horizon Discovery | cat# HD R02-019; RRID:CVCL_HD76 | |
| Recombinant DNA reagent | psiCHECK-2+PTEN3'UTR | Addgene | plasmid# 50936; RRID:Addgene_50936 | |
| Recombinant DNA reagent | *SERINC1* 3'UTR | doi:10.1016/j.cell.2011.09.029 | | Shared by Dr. Pier Paolo Pandolfi, Beth Israel Deaconess Medical Center |
| Recombinant DNA reagent | *VAPA* 3'UTR1 | doi:10.1016/j.cell.2011.09.029 | | Shared by Dr. Pier Paolo Pandolfi, Beth Israel Deaconess Medical Center |
| Recombinant DNA reagent | *VAPA* 3'UTR2 | doi:10.1016/j.cell.2011.09.029 | | Shared by Dr. Pier Paolo Pandolfi, Beth Israel Deaconess Medical Center |
| Recombinant DNA reagent | *CNOT6L* 3'UTR1 | doi:10.1016/j.cell.2011.09.029 | | Shared by Dr. Pier Paolo Pandolfi, Beth Israel Deaconess Medical Center |
| Recombinant DNA reagent | *CNOT6L* 3'UTR2 | doi:10.1016/j.cell.2011.09.029 | | Shared by Dr. Pier Paolo Pandolfi, Beth Israel Deaconess Medical Center |
| Recombinant DNA reagent | *PTEN* 3'UTR | doi:10.1016/j.cell.2011.09.029 | | Shared by Dr. Pier Paolo Pandolfi, Beth Israel Deaconess Medical Center |

*Continued on next page*

*Continued*

| Reagent type (species) or resource | Designation | Source or reference | Identifiers | Additional information |
|---|---|---|---|---|
| Sequence-based reagent | siGlo RISC-free siRNA | Dharmacon | cat#:D-001600–01 | |
| Sequence-based reagent | siGENOME non-targeting siRNA | Dharmacon | cat#:D-001210–02 | |
| Sequence-based reagent | siGENOME *SERINC1* | Dharmacon | cat# M-010725–00 | |
| Sequence-based reagent | siGENOME *ZNF460* | Dharmacon | cat# M-032012–01 | |
| Sequence-based reagent | siGENOME *VAPA* | Dharmacon | cat# M-021382–01 | |
| Sequence-based reagent | siGENOME *CNOT6L* | Dharmacon | cat# M-016411–01 | |
| Sequence-based reagent | siGENOME *PTEN* | Dharmacon | M-003023–02 | |
| Sequence-based reagent | TaqMan probe *SERINC1* | Thermo Fisher Scientific | Hs00380375_m1 | |
| Sequence-based reagent | TaqMan probe *ZNF460* | Thermo Fisher Scientific | Hs01104252_m1 | |
| Sequence-based reagent | TaqMan probe *VAPA* | Thermo Fisher Scientific | Hs00427749_m1 | |
| Sequence-based reagent | TaqMan probe *CNOT6L* | Thermo Fisher Scientific | Hs00375913_m1 | |
| Sequence-based reagent | TaqMan probe *PTEN* | Thermo Fisher Scientific | Hs02621230_s1 | |
| Sequence-based reagent | TaqMan probe *PARD3* | Thermo Fisher Scientific | Hs00969077_m1 | |
| Antibody | rabbit anti-PTEN | Cell Signaling Technology | cat#:9559; clone:138G5; RRID:AB_390810 | 1:1000 dilution |
| Antibody | mouse anti-HSP90 | BD Biosciences | cat#:610419; clone:68; RRID:AB_397798 | 1:1000 dilution |
| Antibody | HRP-conjugated donkey anti-rabbit | GE Healthcare | cat#:NA934; RRID:AB_772206 | 1:2000 dilution |
| Antibody | HRP-conjugated rabbit anti-mouse | Abcam | cat#:ab6728; RRID:AB_955440 | 1:2000 dilution |
| Software, algorithm | Veritas Microplate Luminometer software | Turner BioSystems | part#:998–9100; RRID:SCR_018534 | |
| Software, algorithm | StepOne Plus Real-Time PCR software | Applied Biosystems | RRID:SCR_014281 | Version 2.3 |
| Software, algorithm | ImageJ | doi:10.1038/nmeth.2089 | RRID:SCR_003070 | Version 1.50a |
| Software, algorithm | Gen5 software | BioTek Instruments | RRID:SCR_017317 | Version 2.05.5 |
| Software, algorithm | R Project for statistical computing | https://www.r-project.org | RRID:SCR_001905 | Version 3.5.1 |

## Key resources table

As described in the Registered Report (*Phelps et al., 2016*), we attempted a replication of the experiments reported in Figures 3C–D, G–H and 5B, and Supplemental Figures S3A-B of *Tay et al., 2011*. A detailed description of all protocols can be found in the Registered Report (*Phelps et al., 2016*) and are described below with additional information not listed in the Registered Report, but needed during experimentation.

## Cell culture

DU145 cells (ATCC, cat# HTB-81, RRID:CVCL_0105) were grown in DMEM supplemented with 10% Fetal Bovine Serum (FBS) and wild-type HCT116 and DICER$^{Ex5}$ HCT116 cells (Horizon Discovery, cat# HD R02-019, RRID:CVCL_HD76) were grown in DMEM supplemented with 10% FBS. Media was supplemented with 100 U/ml penicillin, 100 µg/ml streptomycin, and 2 mM glutamine and cells were maintained at 37°C in a humidified atmosphere at 5% $CO_2$. Cells were confirmed to be free of mycoplasma contamination as well as confirmed to be the indicated cells lines by STR DNA profiling. Mycoplasma and STR profile tests were performed by DDC Medical (Fairfield, Ohio).

## Transfections

DU145 cells were seeded at $1.2 \times 10^5$ cells per well in a 12-well plates for the *PTEN*-3'UTR luciferase assays. 24 hr later cells were transfected using Lipofectamine 2000 (Life Technologies, cat# 11668500) according to manufacturer's instructions. Briefly, 2 µl Lipofectamine 2000 was mixed with Opti-MEM so the total volume was 100 µl and incubated for 10 min at RT. 100 ng of psiCHECK-2-+PTEN3'UTR (Addgene, plasmid# 50936; RRID:Addgene_50936) and 100 pmol of siRNA or 1 µg of 3'UTR plasmid were brought to a total volume of 100 µl with Opti-MEM. The two solutions were gently mixed and incubated for an additional 20 min at RT. The transfection mixture was added to the appropriate well of cells and incubated at 37°C in a humidified atmosphere at 5% $CO_2$. After 4 hr, growth medium was replaced. Cells were incubated for 72 hr at 37°C in a humidified atmosphere at 5% $CO_2$ until harvested.

DU145, wild-type HCT116, and HCT116 DICER$^{Ex5}$ cells were seeded at $1.3 \times 10^5$ cells per well in 12-well plates and grown overnight for the Western blot and cell growth assays. The next day cells were transfected using Dharmafect 1 (Thermo Fisher Scientific, cat# T200104) according to manufacturer's instructions with 100 nM siRNA. The transfection mixture was added to the appropriate well of cells and incubated at 37°C in a humidified atmosphere at 5% $CO_2$. For the Western blot assay, growth medium was replaced after 4 hr. Cells were incubated for 72 hr at 37°C in a humidified atmosphere at 5% $CO_2$ until harvested. For the Crystal violet proliferation assay, cells were trypsinized after 8 hr and seeded in new plates.

Transfection efficiency was determined with a siGLO RISC-Free transfection control condition 48 hr after transfection by fluorescence microscopy, which was confirmed to be >90% for all experiments reported. siRNA reagents: siGLO RISC-free (Dharmacon, cat# D-001600–01), siGenome non-targeting control 2 (Dharmacon, cat# D-001210–02), siGenome *SERINC1* (Dharmacon, cat# M-010725–00), siGenome *ZNF460* (Dharmacon, cat# M-032012–01), siGenome *VAPA* (Dharmacon, cat# M-021382–01), siGenome *CNOT6L* (Dharmacon, cat# M-016411–01), and siGenome *PTEN* (Dharmacon, cat# M-003023–02). 3'UTR plasmids: pcDNA-*SERINC1*-3'UTR, pcDNA-*VAPA*-3'UTR1, pcDNA-*VAPA*-3'UTR2, pcDNA-*CNOT6L*-3'UTR1, pcDNA-*CNOT6L*-3'UTR2, pCMV-MCS-*PTEN*-3'UTR (shared by Dr. Pier Paolo Pandolfi, Beth Israel Deaconess Medical Center).

## Luciferase assay

Cells were washed with ice-cold PBS, aspirated, and 100 µl 1X lysis buffer was added to cells and placed on an orbital shaker for 10 min. Dissociated cell lysate was gently pipetted to mix and 20 µl of each lysate was added to a well of a white-walled 96-well plate to measure luciferase activity using the dual luciferase reporter assay (Promega, cat# E1960), Veritas Microplate Luminometer (Turner BioSystems, part# 998–9100), and Veritas software (Turner BioSystems; RRID:SCR_018534), according to manufacturer's instructions.

## Quantitative PCR

Total RNA was extracted with TRIzol reagent (Life Technologies, cat# 15596026) and purified with RNeasy kit (Qiagen, cat# 74104) according to manufacturer's instructions. RNA concentration and purity was determined (quality control data available on OSF project of each experiment - see Figure legends). Total RNA (1 µg) was reverse transcribed into cDNA using High Capacity cDNA Archive kit (Life Technologies, cat# 4368814) according to manufacturer's instructions. Reactions consisted of cDNA (4.5 µl of 10X dilution), TaqMan Fast Advanced Mastermix (Life Technologies, cat# 4444964), and probes (TaqMan probes listed in Registered Report; *Phelps et al., 2016*). Reactions were performed in technical triplicate. Cycling conditions were: one cycle: 50°C for 2 min – one cycle: 95°C

for 20 s – 40 cycles: [95°C for 1 min, 60°C for 20 s] using a StepOne Plus Real-Time PCR system (Applied Biosystems) and StepOne software (RRID:SCR_014281), version 2.3. Negative controls containing no cDNA template were included. Relative expression levels were determined using the ΔΔCt method.

## Western blot

Cells were washed with ice-cold PBS, aspirated, and incubated on ice for 20 min with 30 µl lysis buffer (50 mM Tris-HCl pH 7.4, 150 mM NaCl, 1% NP-40, 0.5% sodium deoxycholate, 0.1% SDS, 5 mM EDTA) supplemented with protease inhibitors (Roche Diagnostics, cat# 11873580001) at manufacturer recommended concentrations. Lysed cells were centrifuged at 12,100x*g* for 15 min at 4°C before protein concentration of supernatant was quantified using a Bradford assay following manufacturer's instructions. Lysate samples (5 to 40 µg) were separated by 4–20% Mini-PROTEAN TGX precast protein gels (BioRad, cat# 456–1094) in 1X Tris Glycine SDS-PAGE gel electrophoresis buffer (National Diagnostic, cat# EC-870–4L) according to manufacturer's instructions and then transferred to a nitrocellulose membrane as described in the Registered Report (Protocol 3; *Phelps et al., 2016*). Transfer was confirmed by Ponceau S staining and membranes were blocked with 5% non-fat dry milk in 1X TBS with 0.1% Tween-20 (TBST) for 1–2 hr at room temperature. Membranes were probed with the following primary antibodies diluted in 5% non-fat dry milk in TBST overnight at 4°C: rabbit anti-PTEN [clone 138G5] (Cell Signaling Technology, cat# 9559, RRID:AB_390810), 1:1000 dilution; mouse anti-HSP90 [clone 68] (BD Biosciences, cat# 610418, RRID:AB_397798), 1:1000 dilution. Membranes were washed with TBST and incubated with secondary antibody diluted in 5% non-fat dry milk in TBST for 1 hr at room temperature: HRP-conjugated donkey anti-rabbit (GE Healthcare, cat# NA934, RRID:AB_772206), 1:2000 dilution; HRP-conjugated rabbit anti-mouse (Abcam, cat# ab6728, RRID:AB_955440), 1:2000 dilution. Membranes were washed with TBST and incubated with ECL reagent to visualize signals. Scanned Western blots were quantified using ImageJ software (RRID:SCR_003070), version 1.50a (*Schneider et al., 2012*). Additional methods and data, including full Western blot images, are available on OSF project of each experiment - see Figure legends.

## Crystal violet proliferation assay

Transfected cells were plated at 20,000 cells per well in 12-well plates (enough for 4 days of measurements) with growth medium. Starting the day after plating (designated day 0), every 24 hr the crystal violet assay was performed as described in the Registered Report (Protocol 4; *Phelps et al., 2016*). Absorbance (OD$_{590}$) was measured with a Synergy HT Multi-Mode microplate reader (BioTek Instruments) and Gen5 software (BioTek Instruments; RRID:SCR_017317), version 2.05.5. For each independent biological repeat, OD$_{590}$ for each condition was normalized by dividing the OD$_{590}$ of each day to the OD$_{590}$ for day 0 for that condition to calculate relative OD$_{590}$. AUC was calculated for each condition of each biological repeat. Data files are available on OSF project - see Figure legends.

## Statistical analysis

Statistical analysis was performed with R software (RRID:SCR_001905), version 3.5.1 (*R Development Core Team, 2018*). All data, csv files, and analysis scripts are available on the OSF (https://osf.io/oblj1/). Confirmatory statistical analysis was pre-registered (https://osf.io/f7yjp/) before the experimental work began as outlined in the Registered Report (*Phelps et al., 2016*). Data were checked to ensure assumptions of statistical tests were met (i.e., Levene's test to test for equality of variances, Shapiro-Wilk test for normality). When described in the results, the Bonferroni correction, to account for multiple testings, was applied to the alpha error or the *p*-value. The Bonferroni corrected value was determined by dividing the uncorrected value (0.05) by the number of tests performed. A meta-analysis of a common original and replication effect size was performed with a random effects model and the *metafor* R package (*Viechtbauer, 2010*) (https://osf.io/xgrqp/). The summary data (mean and standard deviation) pertaining to Figures 3C–D, H and 5B, and Supplemental Figures S3A-B of *Tay et al., 2011* were shared by the original authors. The summary data was published in the Registered Report (*Phelps et al., 2016*) and used in the power calculations to determine the sample sizes for this study.

## Data availability

Additional detailed experimental notes, data, and analysis are available on OSF (RRID:SCR_003238; https://osf.io/oblj1/; *Wang et al., 2020*). This includes the R Markdown file (https://osf.io/e2cun/) that was used to compose this manuscript, which is a reproducible document linking the results in the article directly to the data and code that produced them (*Hartgerink, 2017*).

## Deviations from registered report

The fifth protocol of the Registered Report, which was to test if knockdown of ceRNAs resulted in AKT activation in DU145, wild-type HCT116, and Dicer^Ex5 HCT116 cells was conducted, but the results obtained were inconclusive. The Western blot results with the phospho-AKT antibody were inconsistent and largely absent of a signal. This was not due to the Western blot technique as other antibodies gave reliable singles, as illustrated in *Figure 1B*. Instead, the result is likely due to the lack of phosphatase inhibitors included in the lysis buffer. Of note, the original study also did not use phosphatase inhibitors. Since the primary outcome was not able to be observed, the experiment was not included in this manuscript. Results are available at https://osf.io/ju597/. Additional materials and instrumentation not listed in the Registered Report, but needed during experimentation are also listed above.

# Acknowledgements

The Reproducibility Project: Cancer Biology thank Letizia Longo and Pier Paolo Pandolfi (Beth Israel Deaconess Cancer Center) for sharing critical protocol information, data, and reagents, specifically the Luc-*PTEN*-3'UTR and plasmids to express the 3'UTR of the putative *PTEN* ceRNAs. We also thank the following companies for generously donating reagents to the Reproducibility Project: Cancer Biology; American Type and Tissue Collection (ATCC), Applied Biological Materials, BioLegend, Charles River Laboratories, Corning Incorporated, DDC Medical, EMD Millipore, Harlan Laboratories, LI-COR Biosciences, Mirus Bio, Novus Biologicals, Sigma-Aldrich, and System Biosciences (SBI). All experimental work for this replication study was completed within the Pharmacoanalytical Shared Resource in The Ohio State University Comprehensive Cancer Center.

# Additional information

### Group author details

**Reproducibility Project: Cancer Biology**
**Elizabeth Iorns: Science Exchange, Palo Alto, United States; Rachel Tsui: Science Exchange, Palo Alto, United States; Alexandria Denis: Center for Open Science, Charlottesville, United States; Nicole Perfito: Science Exchange, Palo Alto, United States; Timothy M Errington: Center for Open Science, Charlottesville, United States**

### Competing interests

Hongyan Wang, Hanna S Radomska, Mitch A Phelps: PhASR is a Science Exchange associated lab. Reproducibility Project: Cancer Biology: EI, RT, NP: Employed by and hold shares in Science Exchange Inc.The other authors declare that no competing interests exist.

### Funding

| Funder | Author |
| --- | --- |
| Laura and John Arnold Foundation | Reproducibility Project: Cancer Biology |

The Reproducibility Project: Cancer Biology is funded by the Laura and John Arnold Foundation, provided to the Center for Open Science in collaboration with Science Exchange. The funder had no role in study design, data collection and interpretation, or the decision to submit the work for publication.

## Author contributions
Hongyan Wang, Hanna S Radomska, Data curation, Methodology, Writing - review and editing; Mitch A Phelps, Supervision, Writing - review and editing; Reproducibility Project: Cancer Biology, Formal analysis, Funding acquisition, Visualization, Writing - original draft, Project administration, Writing - review and editing

## Author ORCIDs

Timothy M Errington http://orcid.org/0000-0002-4959-5143

## Decision letter and Author response
Decision letter https://doi.org/10.7554/eLife.56651.sa1
Author response https://doi.org/10.7554/eLife.56651.sa2

# Additional files
## Supplementary files
• Transparent reporting form

## Data availability
Additional detailed experimental notes, data, and analysis are available on OSF (RRID:SCR_003238) (https://osf.io/oblj1/; Wang et al., 2020). This includes the R Markdown file (https://osf.io/e2cun/) that was used to compose this manuscript, which is a reproducible document linking the results in the article directly to the data and code that produced them (Hartgerink, 2017).

The following dataset was generated:

| Author(s) | Year | Dataset title | Dataset URL | Database and Identifier |
|---|---|---|---|---|
| Wang H, Radomska HS, Phelps MA, Iorns E, Tsui R, Denis A, Perfito N, Errington TM | 2020 | Study 24: Replication of Tay et al., 2011 (Cell) | https://doi.org/10.17605/OSF.IO/OBLJ1 | Open Science Framework, 10.17605/OSF.IO/OBLJ1 |

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
