## [Decision Letter]

Thank you for submitting your article "Replication Study: Coding-Independent Regulation of the Tumor Suppressor PTEN by Competing Endogenous mRNAs" for consideration by *eLife*. Your article has been reviewed by three peer reviewers, and the evaluation has been overseen by Timothy Nilsen as Reviewing Editor and Anna Akhmanova as the Senior Editor. The reviewers have opted to remain anonymous.

The reviewers have discussed the reviews with one another and the Reviewing Editor has drafted this decision to help you prepare a revised submission.

The reviewers concurred that the paper presented a sincere and thorough attempt to replicate the results of "Coding-independent regulation of the tumor suppressor PTEN by competing endogenous mRNAs" (Tay et al., 2011). A major conclusion is that Errington and colleagues were not able to reproduce the results of the Tay et al., 2011 paper. Both of the reviewers also questioned the validity of the ceRNA hypothesis, a hypothesis that serves as the foundation for Tay et al., 2011. The overall sentiment was to accept the paper provided that the authors revise the manuscript to address the points raised by reviewers 1 and 2.

Reviewer #1:

The present paper reports the results of a Replication Study detailed in the Registered Report (Phelps et al., 2016). The authors tried to replicate a few selected experiments from the paper "Coding-independent regulation of the tumor suppressor PTEN by 45 competing endogenous mRNAs" (Tay et al., 2011), and compare the outcomes of the original experiments and the replications.

This paper reports that this replication study does not support the findings in the original study for several replicated experiments.

For example, the replication study demonstrated that depletion of putative PTEN competing endogenous mRNAs (ceRNAs) in DU145 cells did not impact PTEN 3'UTR regulation using a reporter, while the original study reported decreased activity when SERINC1, VAPA, and CNOT6L were depleted. The authors suggest that differences between the original study and this replication attempt, such as level of knockdown efficiency and cellular confluence, are factors that might have influenced the results. Specifically, the average reduction in gene expression relative to control siRNA in the replication study was 65% when SERINC1, VAPA, CNOT6L, or PTEN was targeted, but was only 21% for ZNF4. In contrast, the original study reported an achieved knockdown of 90% or greater when SERINC1, VAPA, CNOT6L, or PTEN was targeted, but was 65% for ZNF460 (Tay et al., 2011). Since the purpose of the Replication Study is to replicate the original study, are there reasons or technical difficulties that prevent the authors from conducting the experiments with the same level of knockdown of SERINC1, VAPA, CNOT6L, PTEN, and ZNF460?

In addition, the replication study showed that over-expression of ceRNA 3'UTRs resulted in decreased activity, while the original study reported increased activity. Further, the replication study found depletion of the ceRNAs VAPA or CNOT6L did not statistically impact DU145, wild-type HCT116, or Dicer^Ex5^ HCT116 cell proliferation, while the original study reported increased DU145 and wild-type HCT116 cell proliferation when these ceRNAs were depleted, which was attenuated in the Dicer^Ex5^ HCT116 cells. I assume that sample size of each replication experiment was determined with the variance estimated from the original study. I am wondering whether there is any significant difference in variances between the original study and the replication study and whether this difference can partially explain the discrepancy between the original study and the replication study.

Reviewer #2:

As part of the Reproducibility Project, the authors attempt to replicate selected experiments from Tay et al., 2011, who interpreted their results as evidence that PTEN is regulated through a mechanism involving competing endogenous mRNAs (ceRNAs). Although most of the results of Tay et al. do not appear to be reproducible, the authors of the current paper appear to have done what they could do to rigorously repeat the earlier experiments.

1) The Introduction provides many examples of other putative ceRNAs that have been proposed to regulate PTEN in the past five years (Introduction, third paragraph). However, no mention is made of either the controversy regarding the ceRNA hypothesis or the fact that experiments designed to test the feasibility of the ceRNA hypothesis have indicated that these proposed ceRNAs are expressed at levels that are far too low to enable their physiological variability in expression to impact expression of PTEN or other miRNA targets through the proposed ceRNA mechanism [see Denzler et al., 2016, and references therein]. The issue is that for each miRNA the number of target sites in the transcriptome is very high (typically estimated to exceed 50,000 sites per cell), and thus a physiologically feasible change in the level of a putative ceRNA has little hope of perceptively influencing the regulation other targets--it's just not possible because each putative ceRNA contributes such a small fraction of the total sites for each miRNA. Listing these numerous examples of putative ceRNAs proposed in the last five years without also mentioning either the controversy or the data calling into question the ceRNA hypothesis might mislead readers into thinking that the ceRNA hypothesis advocated by Tay et al., 2011, is plausible and generally accepted by the miRNA research community.

2) When discussing the results of experiment #1, the authors suggested that their inability to reproduce the results of the original study might be attributable to their inability to match the knockdown efficiencies achieved in the original study. They were able to knockdown the putative ceRNAs with an average efficiency of 65%, whereas the original study reported an average knockdown efficiency of 90%. When knocking down these putative ceRNAs, it would have indeed been better to have been able to reproduce the knockdown efficiencies of the original paper, and if the goal of the experiment was to evaluate the function of the protein products of these mRNAs, then the weaker knockdowns would have been a major concern, since a 65% knockdown is often not sufficient to observe reduced protein function. However, when evaluating the putative ceRNA function of these mRNAs, the weaker knockdown is less of a concern. Here, a 65% knockdown should capture 65% of the ceRNA effect, which corresponds to nearly three quarters of the effect expected with a 90% knockdown (0.65/0.90 = .72). Thus, the inability to detect any hint of a ceRNA effect in an experiment in which most of the effect was expected is informative. The original paper interpreted the results of these knockdowns from the perspective of the ceRNA hypothesis, and the current paper would be more informative if it did the same.

3) The summary of the results of the experiments with Dicer^Ex5^ HCT116 cells (Abstract and subsection “ceRNA depletion on PTEN expression”, last paragraph) does not convey the large difference between the results observed in the two studies. In this experiment, it is not the direction of the change that matters--it's the magnitude of the change that matters. The original study observed small differences (< 1.2-fold) when performing the knockdown of putative ceRNAs in Dicer^Ex5^ HCT116 cells, which were interpreted as evidence of an attenuated ceRNA effect. The attempt to repeat this result found large differences (1.4- to 2.9-fold), which cannot be simply interpreted as an attenuated ceRNA effect. Although this discrepancy between the two studies is correctly summarized later (subsection “Meta-analyses of original and replication effects”, fourth paragraph), it is not clearly stated when it is first described (subsection “ceRNA depletion on PTEN expression”, last paragraph) or summarized in the Abstract. Thus, as with the results of experiment #1, the results of this experiment would be more informative if viewed from the perspective of the ceRNA hypothesis. Although it would be beyond the scope of this paper to come up with new hypotheses to explain any results that differ from those of the original paper, it would be appropriate to state whether the results of the replication attempt support the ceRNA hypothesis put forward in the original paper.

---

## [Author Response]

Reviewer #1:The present paper reports the results of a Replication Study detailed in the Registered Report (Phelps et al., 2016). The authors tried to replicate a few selected experiments from the paper "Coding-independent regulation of the tumor suppressor PTEN by 45 competing endogenous mRNAs" (Tay et al., 2011), and compare the outcomes of the original experiments and the replications.This paper reports that this replication study does not support the findings in the original study for several replicated experiments.For example, the replication study demonstrated that depletion of putative PTEN competing endogenous mRNAs (ceRNAs) in DU145 cells did not impact PTEN 3'UTR regulation using a reporter, while the original study reported decreased activity when SERINC1, VAPA, and CNOT6L were depleted. The authors suggest that differences between the original study and this replication attempt, such as level of knockdown efficiency and cellular confluence, are factors that might have influenced the results. Specifically, the average reduction in gene expression relative to control siRNA in the replication study was 65% when SERINC1, VAPA, CNOT6L, or PTEN was targeted, but was only 21% for ZNF4. In contrast, the original study reported an achieved knockdown of 90% or greater when SERINC1, VAPA, CNOT6L, or PTEN was targeted, but was 65% for ZNF460 (Tay et al., 2011). Since the purpose of the Replication Study is to replicate the original study, are there reasons or technical difficulties that prevent the authors from conducting the experiments with the same level of knockdown of SERINC1, VAPA, CNOT6L, PTEN, and ZNF460?

The criteria we used to mirror the original study was the transfection efficiency (>90%), which we were able to achieve using the same conditions as the original study. These criteria were stated in the Registered Report (e.g., Protocol 1, step g. 1.). As with other Replication studies, this is one of many differences that can be considered when interpreting the results of the replication to the original study. However, as raised by reviewer #2 the concern is larger with experiments evaluating protein function. So it is possible, but we did not explore, what conditions would be necessary to achieve a level of knockdown as reported in the original study, while maintaining a healthy cell culture.

In addition, the replication study showed that over-expression of ceRNA 3'UTRs resulted in decreased activity, while the original study reported increased activity. Further, the replication study found depletion of the ceRNAs VAPA or CNOT6L did not statistically impact DU145, wild-type HCT116, or Dicer^Ex5^ HCT116 cell proliferation, while the original study reported increased DU145 and wild-type HCT116 cell proliferation when these ceRNAs were depleted, which was attenuated in the Dicer^Ex5^ HCT116 cells. I assume that sample size of each replication experiment was determined with the variance estimated from the original study. I am wondering whether there is any significant difference in variances between the original study and the replication study and whether this difference can partially explain the discrepancy between the original study and the replication study.

This replication attempt, like all of the replication attempts in the Reproducibility Project: Cancer Biology, were designed to perform independent replications with a calculated sample size to detect the originally reported effect size with at least 80% power based on the originally reported data, which included the means, standard deviations, and sample sizes provided by the original authors. Specifically for the questions raised in this comment: For overexpression of ceRNA 3’UTRs we observed larger RSDs than the original (original: 6-13%; replication: 12-17%), but likely not large enough to fully explain the difference between the original and replication studies. For the cell proliferation experiment, we observed similar RSDs in wild-type and Dicer^Ex5^ HCT116 cells, especially for VAPA and CNOT6L (original: wild-type: VAPA = 8%, CNOT6L = 8%; Dicer^Ex5^: VAPA = 15%, CNOT6L = 6%; replication: wild-type: VAPA = 5%, CNOT6L = 10%; Dicer^Ex5^: VAPA = 12%, CNOT6L = 19%). But for DU145, while the control siRNA conditions were similar between the studies, the original RSDs were much lower in the original study compared to the replication, so is a factor that could influence if statistical significance is reached. In the revised manuscript we included the relative standard deviations (RSDs) for both the original and replication results for all experiments to provide a direct comparison as well as the impact this could have on whether statistical significance is reached.

Reviewer #2:As part of the Reproducibility Project, the authors attempt to replicate selected experiments from Tay et al., 2011, who interpreted their results as evidence that PTEN is regulated through a mechanism involving competing endogenous mRNAs (ceRNAs). Although most of the results of Tay et al. do not appear to be reproducible, the authors of the current paper appear to have done what they could do to rigorously repeat the earlier experiments.1) The Introduction provides many examples of other putative ceRNAs that have been proposed to regulate PTEN in the past five years (Introduction, third paragraph). However, no mention is made of either the controversy regarding the ceRNA hypothesis or the fact that experiments designed to test the feasibility of the ceRNA hypothesis have indicated that these proposed ceRNAs are expressed at levels that are far too low to enable their physiological variability in expression to impact expression of PTEN or other miRNA targets through the proposed ceRNA mechanism [see Denzler et al., 2016, and references therein]. The issue is that for each miRNA the number of target sites in the transcriptome is very high (typically estimated to exceed 50,000 sites per cell), and thus a physiologically feasible change in the level of a putative ceRNA has little hope of perceptively influencing the regulation other targets--it's just not possible because each putative ceRNA contributes such a small fraction of the total sites for each miRNA. Listing these numerous examples of putative ceRNAs proposed in the last five years without also mentioning either the controversy or the data calling into question the ceRNA hypothesis might mislead readers into thinking that the ceRNA hypothesis advocated by Tay et al., 2011, is plausible and generally accepted by the miRNA research community.

We have included additional references in the revised manuscript to present a more balanced view of the ceRNA hypothesis.

2) When discussing the results of experiment #1, the authors suggested that their inability to reproduce the results of the original study might be attributable to their inability to match the knockdown efficiencies achieved in the original study. They were able to knockdown the putative ceRNAs with an average efficiency of 65%, whereas the original study reported an average knockdown efficiency of 90%. When knocking down these putative ceRNAs, it would have indeed been better to have been able to reproduce the knockdown efficiencies of the original paper, and if the goal of the experiment was to evaluate the function of the protein products of these mRNAs, then the weaker knockdowns would have been a major concern, since a 65% knockdown is often not sufficient to observe reduced protein function. However, when evaluating the putative ceRNA function of these mRNAs, the weaker knockdown is less of a concern. Here, a 65% knockdown should capture 65% of the ceRNA effect, which corresponds to nearly three quarters of the effect expected with a 90% knockdown (0.65/0.90 = .72). Thus, the inability to detect any hint of a ceRNA effect in an experiment in which most of the effect was expected is informative. The original paper interpreted the results of these knockdowns from the perspective of the ceRNA hypothesis, and the current paper would be more informative if it did the same.

We thank the reviewer for this suggestion and have included this in the discussion of the results of experiment #1.

3) The summary of the results of the experiments with Dicer^Ex5^ HCT116 cells (Abstract and subsection “ceRNA depletion on PTEN expression”, last paragraph) does not convey the large difference between the results observed in the two studies. In this experiment, it is not the direction of the change that matters--it's the magnitude of the change that matters. The original study observed small differences (< 1.2-fold) when performing the knockdown of putative ceRNAs in Dicer^Ex5^ HCT116 cells, which were interpreted as evidence of an attenuated ceRNA effect. The attempt to repeat this result found large differences (1.4- to 2.9-fold), which cannot be simply interpreted as an attenuated ceRNA effect. Although this discrepancy between the two studies is correctly summarized later (subsection “Meta-analyses of original and replication effects”, fourth paragraph), it is not clearly stated when it is first described (subsection “ceRNA depletion on PTEN expression”, last paragraph) or summarized in the Abstract. Thus, as with the results of experiment #1, the results of this experiment would be more informative if viewed from the perspective of the ceRNA hypothesis. Although it would be beyond the scope of this paper to come up with new hypotheses to explain any results that differ from those of the original paper, it would be appropriate to state whether the results of the replication attempt support the ceRNA hypothesis put forward in the original paper.

We thank the reviewer for this suggestion and have included revisions in the discussion of the PTEN protein expression experiment as well as the Abstract to more accurately reflect how the replication results differ from the original study and the impact this has on the microRNA-dependent manner of the ceRNA hypothesis.